# Myersonian Regression

**Allen Liu**
MIT
cliu568@mit.edu

**Renato Paes Leme**
Google Research
renatoppl@google.com

**Jon Schneider**
Google Research
jschnei@google.com

## Abstract

Motivated by pricing applications in online advertising, we study a variant of linear regression with a discontinuous loss function that we term Myersonian regression. In this variant, we wish to find a linear function $f : \mathbb{R}^d \to \mathbb{R}$ that well approximates a set of points $(x_i, v_i) \in \mathbb{R}^d \times [0, 1]$ in the following sense: we receive a loss of $v_i$ when $f(x_i) > v_i$ and a loss of $v_i - f(x_i)$ when $f(x_i) \leq v_i$. This arises naturally in the economic application of designing a pricing policy for differentiated items (where the loss is the gap between the performance of our policy and the optimal Myerson prices).

We show that Myersonian regression is NP-hard to solve exactly and furthermore that no fully polynomial-time approximation scheme exists for Myersonian regression conditioned on the Exponential Time Hypothesis being true. In contrast to this, we demonstrate a polynomial-time approximation scheme for Myersonian regression that obtains an $\epsilon m$ additive approximation to the optimal possible revenue and can be computed in time $O(\exp(\text{poly}(1/\epsilon))\text{poly}(m, n))$. We show that this algorithm is stable and generalizes well over distributions of samples.

## 1 Introduction

In economics, the Myerson price of a distribution is the price that maximizes the revenue when selling to a buyer whose value is drawn from that distribution. Mathematically, if $F$ is the cdf of the distribution, then the Myerson price is

$$p^* = \text{argmax}_p \, p \cdot (1 - F(p))$$

In many modern applications such as online marketplaces and advertising, the seller doesn't just set one price $p$ but must instead price a variety of differentiated products. In these settings, a seller must design a policy to price items based on their features in order to optimize revenue. Thus, in this paper we study the *contextual learning* version of Myersonian pricing. More formally, we get to observe a *training dataset* $\{(x^t, v^t)\}_{t=1..m}$ representing the bids of a buyer on differentiated products. We will assume that the bids $v^t \in [0, 1]$ come from a truthful auction and hence represent the maximum value a buyer is willing to pay for the product. Each product is represented by a vector of features $x^t \in \mathbb{R}^n$ normalized such that $\|x^t\|_2 \leq 1$. The goal of the learner is to design a policy that suggests a price $\phi(x^t)$ for each product $x^t$ with the goal of maximizing the revenue on the underlying distribution $\mathcal{D}$ from which the pairs $(x^t, v^t)$ are drawn. In practice, one would train a pricing policy on historical bids (training) and apply this policy on future products (testing).

Mathematically, we want to solve

$$\max_{\phi \in \mathcal{P}} \mathbb{E}_{(x,v) \sim \mathcal{D}}[\text{REV}(\phi(x); v)] \tag{PP}$$

where $\mathcal{P}$ is a class of pricing policies and REV is the revenue function (see Figure 1)

$$\text{REV}(p; v) = \max(p, 0) \cdot \mathbf{1}\{p \leq v\}$$

having only access to samples of $\mathcal{D}$.

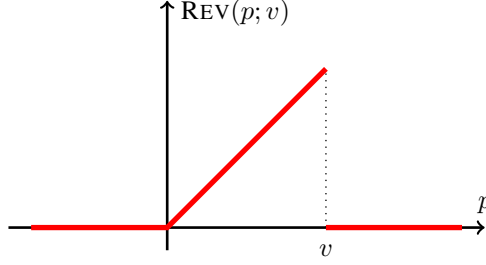

Figure 1: Revenue function

Medina and Mohri [2014a] establish that if the class of policies $\mathcal{P}$ has good generalization properties (defined in terms of Rademacher complexity) then it is enough to solve the problem on the empirical distribution given by the samples. The policy that optimizes over the empirical distribution is typically called *Empirical Risk Minimization* (ERM).

The missing piece in this puzzle is the algorithm, i.e. how to solve the ERM problem. Previous papers (Medina and Mohri [2014a], Medina and Vassilvitskii [2017], Shen et al. [2019]) approached this problem by designing heuristics for ERM and giving conditions on the data under which the heuristics perform well. In this paper we give the first provable approximation algorithm for the ERM problem without assumptions on the data. We also establish hardness of approximation that complements our algorithmic results. We believe these are the first hardness results for this problem. Even establishing whether exactly solving ERM was NP-hard for a reasonable class of pricing policies was open prior to this work.

**Myersonian regression** We now define formally the ERM problem for linear pricing policies[1], which we call *Myersonian regression*. Recall that the dataset is of the form $\{(x^t, v^t)\}_{t=1..m}$ with $x^t \in \mathbb{R}^n$, $\|x^t\|_2 \leq 1$ and $v^t \in [0, 1]$. The goal is to find a linear pricing policy $x \mapsto \langle w, x \rangle$ with $\|w\|_2 \leq 1$ that maximizes the revenue on the dataset, i.e.

$$\max_{w \in \mathbb{R}^n; \|w\|_2 \leq 1} \sum_{t=1}^{m} \text{REV}(\langle w, x^t \rangle; v^t) \tag{MR}$$

It is worth noting that we restrict ourselves to 1-Lipschitz pricing policies by only considering policies with $\|w\|_2 \leq 1$. Bounding the Lipschitz constant of the pricing policy is important to ensure that the problem is stable and hence generalizable. We will contrast it with the unregularized version of (MR) in which the constraint $\|w\|_2 \leq 1$ is omitted:

$$R^* = \max_{w \in \mathbb{R}^n} \sum_{t=1}^{m} \text{REV}(\langle w, x^t \rangle; v^t) \tag{UMR}$$

Without the Lipschitz constraint it is possible to come up with arbitrarily close datasets in the sense that $\|x^t - \tilde{x}^t\| \leq \epsilon$ and $|v^t - \tilde{v}^t| \leq \epsilon$ generating vastly different revenue even as $\epsilon \to 0$. We will also show that (UMR) is APX-hard, i.e. it is NP-hard to approximate within $1 - \epsilon_0$ for some constant $\epsilon_0 > 0$.

**Our Results** Our main result is a polynomial time approximation scheme (PTAS) using dimensionality reduction. We present two versions of the same algorithm.

The first version of the PTAS has running time

$$O(e^{\text{poly}(1/\epsilon)} \cdot \text{poly}(n, m))$$

and outputs an $L$-Lipschitz pricing policy with $L = O(\epsilon\sqrt{n})$ that is an $\epsilon m$-additive approximation of the optimal 1-Lipschitz pricing policy.

The second version of the PTAS has running time

$$O(n^{\text{poly}(1/\epsilon)} \cdot \text{poly}(n, m))$$

and outputs a 1-Lipschitz pricing policy that is an $\epsilon m$-additive approximation of the optimal 1-Lipschitz pricing policy.

We complement this result by showing that the Myersonian regression problem (MR) is NP-hard using a reduction from 1-IN-3-SAT. While it is not surprising that solving Myersonian regression exactly is NP-hard given the discontinuity in the reward function, this has actually been left open by several previous works. In fact, the same reduction implies that under the Exponential Time Hypothesis (ETH) any algorithm approximating it within an $\epsilon m$ additive factor must run in time at least $e^{\Omega(\text{poly}(1/\epsilon))}$, therefore ruling out a fully-polynomial time approximation scheme (FPTAS) for the problem. This hardness of approximation perfectly complements our algorithmic results, showing that our guarantees are essentially the best that one can hope for.

Finally we discuss stability and generalization of the problem. We show that (UMR) is unstable in the sense that arbitrarily small perturbations in the input can lead to completely different solutions. On the other hand (MR) is stable in the sense that the optimal solution varies continuously with the input.

We also discuss the setting in which there is an underlying distribution $\mathcal{D}$ on datapoints $(x, v)$ and while we optimize on samples from $\mathcal{D}$, we care about the loss with respect to the underlying distribution. We also discuss stability of our algorithms and how to extend them to other loss functions. Due to space constraints, most proofs are deferred to the Supplementary Material.

**Related work**   Our work is in the broad area of learning for revenue optimization. The papers in this area can be categorized along two axis: *online* vs *batch* learning and *contextual* vs *non-contextual*. In the online non-contextual setting, Kleinberg and Leighton [2003] give the optimal algorithm for a single buyer which was later extended to optimal reserve pricing in auctions in Cesa-Bianchi et al. [2013]. In the online contextual setting there is a stream of recent work deriving optimal regret bounds for pricing (Amin et al. [2014], Cohen et al. [2016], Javanmard and Nazerzadeh [2016], Javanmard [2017], Lobel et al. [2017], Mao et al. [2018], Leme and Schneider [2018], Shah et al. [2019]). For batch learning in non-contextual settings there is a long line of work establishing tight sample complexity bounds for revenue optimization (Cole and Roughgarden [2014], Morgenstern and Roughgarden [2015, 2016]) as well as approximation algorithms to reserve price optimization (Paes Leme et al. [2016], Roughgarden and Wang [2019], Derakhshan et al. [2019]).

Our paper is in the setting of contextual batch learning. Medina and Mohri [2014a] started the work on this setting by showing generalization bounds via Rademacher complexity. They also observe that the loss function is discontinuous and non-convex and propose the use of a surrogate loss. They bound the difference between the pricing loss and the surrogate loss and design algorithms for minimizing the surrogate loss. Medina and Vassilvitskii [2017] design a pricing algorithm based on clustering, where first features are clustered and then a non-contextual pricing algorithm is used on each cluster. Shen et al. [2019] replaces the pricing loss by a convex loss function derived from the theory of market equilibrium and argue that the clearing price is a good approximation of the optimal price in real datasets. A common theme in the previous papers is to replace the pricing loss by a more amenable loss function and give conditions under which the new loss approximates the pricing loss. Instead here we study the pricing loss directly. We give the first hardness proof in this setting and also give a $(1 - \epsilon)$-approximation without any conditions on the data other than bounded norm.

Our approximation algorithms for this problem works by projecting down to a lower-dimensional linear subspace and solving the problem on this subspace. In this way, it is reminiscent of the area of *compressed learning* (Calderbank et al. [2009]), which studies if it is possible to learn directly in a projected ("compressed") space. More generally, our algorithm fits into a large body of work which leverages the Johnson-Lindenstrauss lemma for designing efficient algorithms (see e.g. Linial et al. [1995] and Har-Peled et al. [2012]).

Hardness of approximation have been established for non-contextual pricing problems with multiple buyers, e.g Paes Leme et al. [2016], Roughgarden and Wang [2019]. Such hardness results hinge on

the interaction between different buyers and don't translate to single-buyer settings. The hardness result in our paper is of a different nature.

## 2   Approximation Algorithms

The main ingredient in the design of our algorithms will be the Johnson-Lindenstrauss lemma:

**Lemma 2.1** (Johnson-Lindenstrauss). *Given a vector $x \in \mathbb{R}^n$ with $\|x\|_2 = 1$, if $\tilde{J}$ is a $k \times n$ matrix formed by taking $k$ random orthogonal vectors as rows for $k = O(\epsilon^{-2} \log \delta^{-1})$ and $J = \sqrt{n/k} \cdot \tilde{J}$, then:*

$$\Pr(|\|Jx\|_2 - 1| > \epsilon) \leq \delta$$

The following is a direct consequence of the JL lemma:

**Lemma 2.2.** *Let $J$ be the JL-projection with $k = O(\epsilon^{-2} \log(1/\epsilon))$, $w^*$ be the optimal solution to (MR) and $x^t$ is a point in the dataset with $\langle w^*, x^t \rangle \geq \epsilon$ then with probability at least $1 - \epsilon$ the following inequalities hold:*

$$(1 - \epsilon) \cdot \|x^t\|_2 \leq \|Jx^t\|_2 \leq (1 + \epsilon) \cdot \|x^t\|_2$$
$$(1 - \epsilon) \cdot \langle w^*, x^t \rangle \leq \langle Jw^*, Jx^t \rangle \leq (1 + \epsilon) \cdot \langle w^*, x^t \rangle$$

**PTAS - Version 1:**   For the first version of the algorithm, we randomly sample $1/\epsilon$ JL-projections $J$ with $k = O(\epsilon^{-2} \log(1/\epsilon))$ and search over an $\epsilon$-net of the projected space. For each projection, we define a set of discretized vectors as:

$$D = \{\hat{w}; \hat{w} = \epsilon^5 z \text{ for } z \in \mathbb{Z}^k, \|\hat{w}\|_2 \leq 1 + \epsilon\}$$

Then we search for the vector $\hat{w} \in D$ that maximizes

$$\sum_{t=1}^{m} \text{REV}(\langle \hat{w}, Jx^t \rangle; v^t) \tag{1}$$

Over all projections, we output the vector $w = J^\top \hat{w}$ that maximizes the revenue.

**Theorem 2.3.** *There is an algorithm with running time $O(e^{\text{poly}(1/\epsilon)} \text{poly}(n, m))$ that outputs a vector $w$ with $\|w\|_2 \leq O(\epsilon \cdot \sqrt{n})$ such that:*

$$\mathbb{E}\left[\sum_t \text{REV}(\langle w, x^t \rangle; v^t)\right] \geq R^* - O(\epsilon m)$$

*where $R^* = \sum_t \text{REV}(\langle w^*, x^t \rangle; v^t)$ for the optimal $w^*$ with $\|w^*\|_2 \leq 1$.*

*Proof.* The running time follows from the fact that $|D| \leq (1/\epsilon)^{O(k)} = e^{O(\text{poly}(1/\epsilon))}$. We show the approximation guarantee in three steps:

*Step 1: defining good points.* Let $w^*$ be the optimal solution to (MR). Say that a datapoint $(x^t, v^t)$ is good if $\epsilon \leq \langle w^*, x^t \rangle \leq v^t$ and the event in Lemma 2.2 happens. If $G$ is the set of indices $t$ corresponding to good datapoints, then with at least $1/2$ probability:

$$\sum_{t \in G} \langle w^*, x^t \rangle \geq R^* - 2\epsilon m$$

This is true since the points with $\langle w^*, x^t \rangle < \epsilon$ can only affect the revenue by at most $\epsilon$ each and for the remaining $m'$ points, each can fail to be good with probability at most $\epsilon$. The revenue loss in expectation is at most $m'\epsilon$, so by Markov's inequality it is at most $2m'\epsilon$ with $1/2$ probability.

*Step 2: projection of the optimal solution.* Define $w' = (1 - 2\epsilon) \cdot Jw^*$ and define $\hat{w}$ to be the vector in $D$ obtained by rounding all coordinates of $w'$ to the nearest multiple of $\epsilon^5$. For any good index $t \in G$ we have:

$$\langle \hat{w}, Jx^t \rangle = \langle \hat{w} - w', Jx^t \rangle + \langle w', Jx^t \rangle \leq (1 + \epsilon)\epsilon^5 \sqrt{k} + (1 - 2\epsilon)\langle Jw^*, Jx^t \rangle$$
$$\leq (1 + \epsilon)\epsilon^5 \sqrt{k} + (1 - \epsilon)\langle w^*, x^t \rangle \leq v^t$$

and hence that datapoint generates revenue since the price is below the value. And:

$$\langle \hat{w}, Jx^t \rangle = \langle \hat{w} - w', Jx^t \rangle + \langle w', Jx^t \rangle \geq -(1 + \epsilon)\epsilon^5 \sqrt{k} + (1 - 2\epsilon)\langle Jw^*, Jx^t \rangle$$
$$\geq -(1 + \epsilon)\epsilon^5 \sqrt{k} + (1 - 5\epsilon)\langle w^*, x^t \rangle$$

*Step 3: bounding the revenue.* Finally, note that

$$\langle w, x^t \rangle = \langle J^\top \hat{w}, x^t \rangle = \langle \hat{w}, Jx^t \rangle$$

so:

$$\sum_t \text{REV}(\langle w, x^t \rangle; v^t) = \sum_{0 \leq \langle \hat{w}, Jx^t \rangle \leq v^t} \langle \hat{w}, Jx^t \rangle \geq \sum_{t \in G} \langle \hat{w}, Jx^t \rangle \geq (1 - 5\epsilon) \sum_{t \in G} \langle w^*, x^t \rangle - O(\epsilon)$$
$$\geq (1 - 5\epsilon)(R^* - 2m\epsilon) - O(\epsilon m) = R^* - O(\epsilon m)$$

Since we sample $1/\epsilon$ independent JL projections and for each, we find an $O(\epsilon m)$ additive approximation with probability at least $1/2$, our algorithm achieves expected revenue $R^* - O(\epsilon m)$, as desired. $\qquad\square$

**PTAS – Version 2**  The main drawback of the first version of the PTAS is that we output an $\epsilon\sqrt{n}$-Lipschitz pricing policy that is an approximation to the optimal 1-Lipschitz pricing policy. With an increase in running time, it is possible to obtain the same approximation with an 1-Lipschitz pricing policy (i.e. $\|w\|_2 \leq 1$). For that we will increase the dimension of the JL projection to $k = O(\epsilon^{-2}\log(n/\epsilon))$. This will allow us to have the following conditions hold simultaneously for all datapoints with probability at least $1 - \epsilon$:

$$(1 - \epsilon) \cdot \|x^t\|_2 \leq \|Jx^t\|_2 \leq (1 + \epsilon) \cdot \|x^t\|_2$$
$$\langle w^*, x^t \rangle - \epsilon^2 \leq \langle Jw^*, Jx^t \rangle \leq \langle w^*, x^t \rangle + \epsilon^2$$

This follows from the same argument in Lemma 2.2, taking the Union Bound over all points. Now we repeat the following process $(1/\epsilon)^{O(k\log(1/\epsilon))}$ times:

Choose a random point $\hat{w}$ in the unit ball in $\mathbb{R}^k$. For each such $\hat{w}$ we define the important set as $t \in \hat{G}(\hat{w})$ if $10\epsilon \leq \langle \hat{w}, Jx^t \rangle \leq v^t$. Now, we check (by solving a convex program) if there exists a vector $w \in \mathbb{R}^n$ with $\|w\|_2 \leq 1$ such that:

$$\frac{\langle \hat{w}, Jx^t \rangle}{1 + 5\epsilon} \leq \langle w, x^t \rangle \leq v^t, \forall t \in \hat{G}(\hat{w})$$

If it exists, call it $w(\hat{w})$ otherwise discard $\hat{w}$. Over all $(1/\epsilon)^{O(k\log(1/\epsilon))}$ iterations, for all vectors $\hat{w}$ that weren't discarded, choose the one maximizing the objective (1) and output $w(\hat{w})$.

**Theorem 2.4.**  There is an algorithm with running time $O(n^{\text{poly}(1/\epsilon)}\text{poly}(n,m))$ that outputs a vector $w$ with $\|w\|_2 \leq 1$ such that:

$$\mathbb{E}\left[\sum_t \text{REV}(\langle w, x^t \rangle; v^t)\right] \geq R^* - O(\epsilon m)$$

where $R^* = \sum_t \text{REV}(\langle w^*, x^t \rangle; v^t)$ for the optimal $w^*$ with $\|w^*\|_2 \leq 1$.

*Proof. Step 1: When $\hat{w}$ lies close to the projection of the optimum, the convex program is feasible*

Let $w' = (1 - 2\epsilon) \cdot Jw^*$. If $\|\hat{w} - w'\| \leq \epsilon^5$ we will show that the convex program is solvable. For $t \in \hat{G}(\hat{w})$ we have

$$\langle w^*, x^t \rangle \leq \frac{1}{1 - 2\epsilon}\langle w', Jx^t \rangle + \epsilon^2 \leq (1 + 3\epsilon)(\langle \hat{w}, Jx^t \rangle + (1 + \epsilon)\epsilon^5) + \epsilon^2 \leq (1 + 5\epsilon)v^t$$

and

$$\langle w^*, x^t \rangle \geq \frac{1}{(1 - 2\epsilon)}\langle w', Jx^t \rangle - \epsilon^2 \geq (1 + 2\epsilon)\langle w', Jx^t \rangle - \epsilon^2$$
$$\geq (1 + 2\epsilon)(\langle \hat{w}, Jx^t \rangle - (1 + \epsilon)\epsilon^5) - \epsilon^2 > \langle \hat{w}, Jx^t \rangle$$

Thus $1/(1 + 5\epsilon) \cdot w^*$ is a solution to the convex program.

*Step 2: When $\hat{w}$ lies close to the projection of the optimum, any solution to the convex program achieves a good approximation*

If $||\hat{w} - w'|| \leq \epsilon^5$ then for each data point $x^t$ with $t \in \hat{G}(\hat{w})$

$$\langle \hat{w}, Jx^t \rangle = \langle \hat{w} - w', Jx^t \rangle + \langle w', Jx^t \rangle \geq -(1+\epsilon)\epsilon^5 + (1 - 2\epsilon)\langle Jw^*, Jx^t \rangle$$
$$\geq -(1+\epsilon)\epsilon^5 + (1 - 5\epsilon)\langle w^*, x^t \rangle$$

Note the last step holds because

$$\langle w^*, x^t \rangle \geq \langle \hat{w}, Jx^t \rangle \geq 10\epsilon$$

and

$$\langle Jw^*, Jx^t \rangle \geq \langle w^*, x^t \rangle - \epsilon^2.$$

Next, we deal with the datapoints with $t \notin \hat{G}(\hat{w})$. For these datapoints, either $\langle \hat{w}, Jx^t \rangle < 10\epsilon$ in which case
$$\langle w^*, x^t \rangle \leq (1 + 5\epsilon)\langle w', Jx^t \rangle + \epsilon^2$$
$$\leq (1 + 5\epsilon)(\langle \hat{w}, Jx^t \rangle + (1+\epsilon)\epsilon^5) + \epsilon^2 \leq 11\epsilon$$

or $\langle \hat{w}, Jx^t \rangle > v^t \geq 10\epsilon$ in which case

$$\langle w^*, x^t \rangle \geq \frac{1}{(1 - 2\epsilon)}\langle w', Jx^t \rangle - \epsilon^2 \geq (1 + 2\epsilon)\langle w', Jx^t \rangle - \epsilon^2$$
$$\geq (1 + 2\epsilon)(\langle \hat{w}, Jx^t \rangle - (1+\epsilon)\epsilon^5) - \epsilon^2 > (1 + 2\epsilon)(v^t - (1+\epsilon)\epsilon^5) - \epsilon^2 > v^t$$

Thus, the total revenue achieved by $w(\hat{w})$ is at least

$$\frac{1}{1 + 5\epsilon} \sum_{t \in \hat{G}(\hat{w})} \left( -2\epsilon^5 + (1 - 5\epsilon)\text{REV}(\langle w^*, x^t \rangle; v^t) \right)$$
$$\geq -2\epsilon^5 m + (1 - 10\epsilon) \sum_{t \in \hat{G}(\hat{w})} \text{REV}(\langle w^*, x^t \rangle; v^t)$$
$$\geq -2\epsilon^5 m + (1 - 10\epsilon) \left( \sum_{t} \text{REV}(\langle w^*, x^t \rangle; v^t) - 11\epsilon m \right)$$
$$\geq \sum_{t} \text{REV}(\langle w^*, x^t \rangle; v^t) - 25\epsilon m$$

*Step 3: The algorithm finds a good approximation with probability $1 - O(\epsilon)$*

It suffices to show that our algorithm will choose some $\hat{w}$ such that $||\hat{w} - w'|| \leq \epsilon^5$ with probability $1 - O(\epsilon)$. Note
$$||w'||_2 \leq (1 - 2\epsilon)(1 + \epsilon) \leq 1 - \epsilon.$$

Thus the probability that $\hat{w}$ lands within distance $\epsilon^5$ of $w'$ is $\epsilon^{5k}$. Since we choose $(1/\epsilon)^{O(k \log(1/\epsilon))}$ different points $\hat{w}$ independently at random, the probability that at least one of them lands within distance $\epsilon^5$ of $w'$ is at least $1 - \epsilon$. □

## 3 Hardness of approximation

Unlike $\ell_2$ and $\ell_1$ regression, Myersonian regression is NP-hard. We prove two hardness results. First we show that without the assumption $||w||_2 \leq 1$, achieving a constant factor approximation is NP-hard. Then we show that under the Exponential Time Hypothesis (ETH), any algorithm that achieves a $\epsilon m$-additive approximation for Myersonian regression must run in time at least $\exp(O\left(\epsilon^{-1/6}\right))$.

**1-in-3-SAT** We will rely on reductions from the 1-IN-3-SAT problem, which is NP-complete. The input to 1-IN-3-SAT is an expression in conjunctive normal form with each expression having 3 literals per clause (i.e. a collection of expression of the type $X_i \vee X_j \vee \overline{X_k}$). The problem is to determine if there is a truth assignment such that exactly one literal in each clause is true (and the remaining are false).

**GAP 1-in-3-SAT**   We will need a slightly stronger hardness result that 1-in-3-SAT is not only hard to solve exactly, but it is hard to approximate the maximum number of clauses that can be satisfied. In particular, there are constants $0 < c_1 < c_2 \leq 1$ such that given a 1-in-3-SAT instance, it is NP-hard to distinguish the following two cases

- At most $c_1$-fraction of the clauses can be satisfied
- At least $c_2$-fraction of the clauses can be satisfied

**ETH**   The Exponential Time Hypothesis says that 3-SAT with $N$ variables can't be solved in time $O(2^{cN} \text{poly}(N))$ for some constant $c > 0$. Since there is a linear time reduction between 3-SAT and 1-IN-3-SAT and 1-IN-3-SAT is NP-complete, then ETH implies that there is no $O(2^{cN} \text{poly}(N))$ time algorithm for 1-IN-3-SAT.

**Lemma 3.1.**  *There exists a constant $\epsilon > 0$ for which it is possible to reduce (in poly-time) an instance of $(c_1, c_2)$-GAP 1-in-3-SAT to computing a $(1 - \epsilon)$-approximation for an instance of the unregularized Myersonian regression problem* (UMR).

**Theorem 3.2.**  There is some constant $\epsilon > 0$ for which obtaining a $(1 - \epsilon)$-approximation for the unregularized Myersonian regression problem (UMR) is NP-hard.

The proof follows directly from Lemma 3.1 and the NP hardness of GAP-1-IN-3-SAT. The previous result rules out a PTAS for (UMR). In contrast we will see that while (MR) is still NP-hard to solve exactly, it admits a PTAS. However, runtime that is superpolynomial in $\epsilon$ is necessary.

**Lemma 3.3.**  *It is possible to transform (in poly-time) an instance of* 1-IN-3-SAT *with $N$ variables into an instance of Myersonian regression with the promise $\|w\|_2 \leq 1$ and $n = O(N)$ and $m = O(N^5)$ in such a way that a satisfiable* 1-IN-3-SAT *instance will map to an instance of Myersonian regression with revenue $R \leq O(N^{2.5})$ while any unsatisfiable instance will map to an instance with revenue at most $R - 0.5N^{-0.5}$.*

If we assume ETH, we obtain a bound on the runtime of any approximation algorithm:

**Theorem 3.4.**  Under ETH, any algorithm that achieves a $\epsilon m$-additive (or $(1 - \epsilon)$-multiplicative) approximation for Myersonian regression must run in time at least $O(2^{\Omega(\epsilon^{-1/6})} \text{poly}(n, m))$.

*Proof.*  Assume there is an approximation algorithm for Myersonian regression with running time $O(2^{\Omega(\epsilon^{-1/6})} \text{poly}(n, m))$ for the constant $c$ in the definition of ETH.

The for an instance of 1-IN-3-SAT with $N$ variables, consider the transformation in Lemma 3.3 and apply the approximation algorithm with $\epsilon = O(1/N^6)$. Such an approximation algorithm would run in time $O(2^{cN} \text{poly}(N))$ and distinguish between the satisfiable and unsatisfiable cases of 1-IN-3-SAT, contradicting ETH.  $\square$

## 4   Stability, Generalization and Extensions

We start by commenting on the importance of the constraint $\|w\|_2 \leq 1$ imposed on the problem (MR), which is closely related to stability and generalization.

**Offset term**   It will be convenient to allow a constant term in the pricing loss, i.e. we will look at pricing functions of the type:

$$x \mapsto w_1 + \sum_{i=2}^{n} w_i x_i^t$$

This is equivalent to assuming that all the datapoints have $x_1^t = 1$ and $\|x^t\|_2 \leq \sqrt{2}$. We renormalize such that we still have $\sum_{i=2}^{n}(x_i^t)^2 \leq 1$. We will make this assumption for the rest of this section.

We note that this assumption doesn't affect the results in the previous sections. The positive results remain unchanged since we don't have any assumption on the data other than the norm being bounded by a constant. Our hardness results can be easily adapted to the setting with an offset term. We can essentially force the constant term to be very small by adding $\Omega(N^{103})$ data points with $v^t = 1/N^{100}, x_1^t = 1$ and all other coordinates 0.

**Stability** We start by discussing the constraint $\|w\|_2 \leq 1$ imposed on the problem (MR). Without this constraint, it is possible to completely change the objective function with a tiny perturbation in the problem data. Let $R^*$ be the optimal revenue in the unregularized Myersonian regression (UMR) for some instance $(x^t, v^t)$. A natural upper bound on $R^*$ is the maximum welfare, given by $W = \sum_{t=1}^m v^t$. Typically $R^* < W$. Consider such an instance. For any fixed $\delta < 0$ consider the following two instances:

- $\tilde{x}^t = (x^t, 0) \in \mathbb{R}^{n+1}$
- $\bar{x}^t = (x^t, \delta v^t) \in \mathbb{R}^{n+1}$

The instances $(\tilde{x}^t, v^t)_{t=1..m}$ and $(\bar{x}^t, v^t)_{t=1..m}$ are very close to each other in the sense that the labels are the same and the features have:
$$\|\tilde{x}^t - \bar{x}^t\| \leq \delta, \forall t.$$
However, the optimal revenue of $(\tilde{x}^t, v^t)_{t=1..m}$ under (UMR) is $R^*$ while the optimal revenue of $(\bar{x}^t, v^t)_{t=1..m}$ is $W$ by choosing $w = (0, \delta^{-1})$. This is true even as $\delta \to 0$.

On the other hand, the solution of the regularized problem (MR) is Lipschitz-continuous in the data.

**Theorem 4.1.** Consider two instances $(\tilde{x}^t, \tilde{v}^t)_{t=1..m}$ and $(\bar{x}^t, \bar{v}^t)_{t=1..m}$ such that $\|\tilde{x}^t - \bar{x}^t\| \leq \delta$ and $|\tilde{v}^t - \bar{v}^t| \leq \delta$ for all $t$, then if $\tilde{R}$ and $\bar{R}$ are the respective solutions to (MR) then:
$$|\tilde{R} - \bar{R}| \leq O(\delta m)$$

**Uniform Convergence and Generalization** To understand generalization, we are concerned with the performance of the algorithm on a distribution $\mathcal{D}$ that generates datapoints $(x^t, v^t)$. We will sample $m$ points from this distribution and obtain a dataset $\mathcal{S} = \{(x^t, v^t); t = 1..m\}$. We want to compare across all pricing policies $w$ the objective function on the sample:

$$F_{\mathcal{S}}(w) = \frac{1}{m} \sum_{t=1}^m \text{REV}(\langle w, x^t \rangle; v^t)$$

with the performance on the original distribution:

$$F_{\mathcal{D}}(w) = \mathbb{E}_{(x,v) \sim \mathcal{D}} \left[ \text{REV}(\langle w, x^t \rangle; v^t) \right]$$

Medina and Mohri [2014a] provide bounds for $|F_{\mathcal{S}}(w) - F_{\mathcal{D}}(w)|$ by studying the empirical Rademacher complexity of the pricing function. The following statement follows directly from Theorem 3 in their paper. Note that while their theorem bounds only one direction, the same proof also works for the other direction.

**Theorem 4.2** (Medina and Mohri [2014a]). For any $\delta > 0$ it holds with probability $1 - \delta$ over the choice of a sample $\mathcal{S}$ of size $m$ that:

$$|F_{\mathcal{S}}(w) - F_{\mathcal{D}}(w)| \leq O\left( \sqrt{\frac{n \log(m/n) + \log(1/\delta)}{m}} \right)$$

**Corollary 4.3.** *Let $w_{\mathcal{S}}$ be the output of the ERM algorithm on sample $\mathcal{S}$ of size $m = O(\epsilon^{-2}[n \log(n/d) + \log(1/\delta)])$. Then with probability $1 - \delta$ we have:*

$$F_{\mathcal{D}}(w_{\mathcal{S}}) \geq \max_{\|w\|_2 \leq 1} F_{\mathcal{D}}(w) - O(\epsilon)$$

**Extensions to other loss functions** While our results are phrased in terms of the pricing, they hold for any lower-semi-Lipschitz reward fuction, i.e. any function such that:
$$R(p - \epsilon) \geq R(p) - \epsilon$$

An important example studied in Medina and Mohri [2014a], Shen et al. [2019] is the revenue of a second price auction with reserves price $p$. Given two highest bids $v_1$ and $v_2$ the revenue function is written as:
$$\text{SPA}(p; v_1, v_2) = \max(v_2, p) \cdot \mathbf{1}\{p \leq v_1\}$$

# 5    Conclusion

We give the first approximation algorithm for learning a linear pricing function without any assumption on the data other than normalization. This provides a key missing component to the field of learning for revenue optimization, where ERM was shown to be optimal in Medina and Mohri [2014a] but there were no algorithms with provable guarantees for it.

Our algorithm is polynomial in the number of features dimensions $n$ and on the number of datapoints $m$ but exponential in the accuracy parameter $\epsilon$. We show that the exponential dependency on $\epsilon$ is necessary.

In this paper we assume that the bids in the dataset represent the buyer's true willingness to pay as in Medina and Mohri [2014a], Medina and Vassilvitskii [2017], Shen et al. [2019]. A interesting avenue of investigation for future work is to understand how strategic buyers would change their bids in response to a contextual batch learning algorithm and how to design algorithms that are aware of strategic response. This is a well studied problem in non-contextual online learning (Amin et al. [2013], Medina and Mohri [2014b], Drutsa [2017], Vanunts and Drutsa [2019], Nedelec et al. [2019]) as well as in online contextual learning (Amin et al. [2014], Golrezaei et al. [2019]). Formulating a model of strategic response to batch learning algorithms is itself open.

### Broader Impact Statement

While our work is largely theoretical, we feel it can have downstream impact in the design of better marketplaces such as those for internet advertisement. Better pricing can increase both the efficiency of the market and the revenue of the platform. The latter is important since the revenue of platforms keeps such services (e.g. online newspapers) free for most users.

## Acknowledgments and Disclosure of Funding

No funding to disclose. The authors would like to thank Andrés Muñoz Medina for helpful discussions.

## Footnotes

[1]The choice of linear function is actually not very restrictive. A common trick in machine learning is to map the features to a different space and train a linear model on $\psi(x)$. For example if $d = 2$, the features are $(x_1, x_2)$. By mapping $\psi(x) = (1, x_1, x_2, x_1^2, x_2^2, x_1 x_2) \in \mathbb{R}^6$, and training a linear function on $\psi(x)$, we are actually optimizing over all quadratic functions on the original features. Similarly, we can optimize over any polynomial of degree $k$ or even more complex functions with an adequate mapping.

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
