[Supplementary Material]

# A Omitted Proofs

**Proof of Lemma 2.1**

*Proof of Lemma 2.1.* Apply the JL lemma for vectors $x^t/\|x^t\|_2$, $w^*/\|w^*\|_2$ and $(x^t + w^*)/\|x^t + w^*\|_2$ with $\delta = \epsilon/3$. Then with probability at least $1 - \epsilon$ the following three inequalities hold (using the Union Bound):

$$(1 - \epsilon) \cdot \|x^t\|_2 \leq \|Jx^t\|_2 \leq (1 + \epsilon) \cdot \|x^t\|_2$$

$$(1 - \epsilon) \cdot \|w^*\|_2 \leq \|Jw^*\|_2 \leq (1 + \epsilon) \cdot \|w^*\|_2$$

$$(1 - \epsilon) \cdot \|x^t + w^*\|_2 \leq \|J(x^t + w^*)\|_2 \leq (1 + \epsilon) \cdot \|x^t + w^*\|_2$$

Since we can write the dot-product as follows:

$$\langle w^*, x^t \rangle = \frac{1}{2} \left( \|w^* + x^t\|_2^2 - \|w^*\|_2^2 - \|x^t\|_2^2 \right)$$

$$\langle Jw^*, Jx^t \rangle = \frac{1}{2} \left( \|Jw^* + Jx^t\|_2^2 - \|Jw^*\|_2^2 - \|Jx^t\|_2^2 \right)$$

then we have:

$$|\langle Jw^*, Jx^t \rangle - \langle w^*, x^t \rangle| \leq O(\epsilon^2) \leq O(\epsilon) \cdot \langle w^*, x^t \rangle$$

$\square$

**Proof of Lemma 3.1**

*Proof of Lemma 3.1.* We proceed in three steps:

*Step 1: define a transformation from* 1-IN-3-SAT *to Myersonian regression.* Consider a 1-IN-3-SAT instance with $N$ variables $X_1, \ldots, X_N$. For $1 \leq i \leq N$ let $s_i$ be the number of clauses that $X_i$ appears in. Let $K$ be a sufficiently large constant (depending only on $c_1, c_2$). We will map to an instance of Myersonian regression with $n = 2N$ variables, where $i = 1..N$ will correspond to boolean literals $X_i$ and $i = (N+1)..2N$ will correspond to negated literals $\overline{X_i}$. We will build the instance as follows: for each $i = 1..N$ we will create the following datapoints $(x^t, v^t)$. In each case, the unset coordinates are zero.

- $K^2 s_i$ datapoints with $v^t = 1 - \frac{2}{K s_i}$, $x_i^t = 1$.

- $K^2 s_i$ datapoints with $v^t = 1 - \frac{2}{K s_i}$, $x_{N+i}^t = 1$.

- $K^3 s_i^2 - K^2 s_i$ datapoints with $v^t = \frac{1}{K s_i}$, $x_i^t = 1$.

- $K^3 s_i^2 - K^2 s_i$ datapoints with $v^t = \frac{1}{K s_i}$, $x_{N+i}^t = 1$.

- $K^2 s_i$ points with $v^t = 1 - \frac{1}{K s_i}$, $x_i^t = 1$, $x_{N+i}^t = 1$

We call these data points auxiliary data points. Now for each clause $X_i \vee \overline{X_j} \vee X_k$ we will add a datapoint with $v^t = 1$ and $x_i^t = x_{N+j}^t = x_k^t = 1$. We call these data points clause-data points. This concludes the transformation[2]

*Step 2: we show that the optimal revenue of the Myersonian regression is attained when for each* $i$ *exactly one of* $w_i, w_{i+N}$ *is in the interval* $\left( \frac{2}{3K s_i}, \frac{1}{K s_i} \right]$ *and the other is in the interval* $\left( \frac{2}{3}, 1 \right]$. *Furthermore, the maximum possible revenue from all auxiliary data points is* $3 \cdot K^2(s_1 + \cdots + s_N) - 3KN$.

First note that if we set $w_i = 1 - \frac{2}{Ks_i}$ and $w_{i+N} = \frac{1}{Ks_i}$ the total revenue from the auxiliary data points involving $w_i$ and $w_{i+N}$ is

$$K^2 s_i \left(1 - \frac{2}{Ks_i}\right) + K^3 s_i^2 \left(\frac{1}{Ks_i}\right) + K^2 s_i \left(1 - \frac{1}{Ks_i}\right)$$
$$= 3 \cdot K^2 s_i - 3K$$

Now we verify that in each of the following cases, if we fix the values of $w_j, w_{j+N}$ for $j \neq i$, then the revenue can be strictly increased by setting $w_i = 1 - \frac{2}{Ks_i}$ and $w_{i+N} = \frac{1}{Ks_i}$:

- $w_i \leq \frac{2}{3Ks_i}$ or $w_{i+N} \leq \frac{2}{3Ks_i}$

  If $w_i \leq \frac{2}{3Ks_i}$ then the total revenue from the auxiliary data points involving $w_i, w_{i+N}$ is at most

  $$K^3 s_i^2 \left(\frac{2}{3Ks_i}\right) + \max\left(K^3 s_i^2 \left(\frac{1}{Ks_i}\right), K^2 s_i \left(1 - \frac{2}{Ks_i}\right)\right) + K^2 s_i \left(1 - \frac{1}{Ks_i}\right)$$

  which is at most $2.7 K^2 s_i$. If we instead set $w_i = 1 - \frac{2}{Ks_i}$ and $w_{i+N} = \frac{1}{Ks_i}$, we increase the revenue from auxiliary data points by at least $0.3K^2 s_i - 3K$ and we affect at most $s_i$ clause data points so the total revenue is increased.

- $\frac{1}{Ks_i} < w_i \leq \frac{2}{3}$ or $\frac{1}{Ks_i} < w_{i+N} \leq \frac{2}{3}$

  This case is dealt with similar to the above.

- $w_i + w_{i+N} \leq \frac{2}{3}$ or $w_i + w_{i+N} > 1 - \frac{1}{Ks_i}$

  This case is dealt with similar to the above.

The main claim in this step can be verified by inspecting the leftover regions, which correspond to the white regions in Figure 2.

Figure 2: Optimal revenue for the instance in the reduction are achieved for $(w_i, w_{i+N})$ in the white region.

*Step 3: Bound the revenue for $c_1$-unsatisfiable and $c_2$-satisfiable* 1-IN-3-SAT *instances.* If the instance is $c_2$-satisfiable, then we can assign $x_i = 1 - \frac{2}{Ks_i}$ and $x_{N+i} = \frac{1}{Ks_i}$ when $X_i$ is true in the $c_2$-satisfying assignment and $x_{i+N} = 1 - \frac{2}{Ks_i}$ and $x_i = \frac{1}{Ks_i}$ otherwise. This achieves a total revenue of

$$R_2 = 3K^2(s_1 + \cdots + s_N) - 3KN + c_2 S$$

where $S$ is the number of clauses in the formula. Note $s_1 + \cdots + s_N = 3S$ so
$$R_2 = (9K^2 + c_2)S - 3KN$$
If the formula is not $c_1$-satisfiable then there can be no solution to the Myersonian regression that achieves revenue more than
$$R_1 = 9K^2 S - 3KN + c_1 S + \frac{3}{K}(1 - c_1)S$$
This is because for any values for the variables, we can consider letting $X_i$ be true in the Boolean formula whenever $w_i \in \left(\frac{2}{3}, 1\right]$ and $X_i$ be false when $w_{i+N} \in \left(\frac{2}{3}, 1\right]$. By assumption, at least $1 - c_1$-fraction of the clauses (e.g. $(X_i \vee \overline{X_j} \vee X_k)$) in the Boolean formula are violated meaning that either there is more than one true literal, in which case:
$$w_i + w_{j+N} + w_k \geq \frac{4}{3}$$
or all literals are false, in which case:
$$w_i + w_{j+N} + w_k \leq \frac{3}{K}$$
Now clearly $S > N/3$ (since each variable must appear in at least one clause). Since $0 < c_1 < c_2 \leq 1$ are fixed constants (independent of $N$), if we choose $K$ sufficiently large in terms of $c_1, c_2$, there is a $(1 - \epsilon)$-factor gap between $R_1$ and $R_2$ for some small constant $\epsilon > 0$ independent of $N$. $\qquad\square$

**Proof of Lemma 3.3**

*Proof of Lemma 3.3.* Note we can assume that in the original 1-IN-3-SAT instance, there are at most $O(N^3)$ clauses and each variable appears in at most $O(N^2)$ clauses. In the instance constructed in the proof of Lemma 3.1, the optimal solution $w$ has $||w||_2 = O(\sqrt{N})$. Construct the same instance but with all values $v^t$ scaled down by a factor of $1/\sqrt{N}$. Call this instance $M$.

Following the same argument as in the proof of Lemma 3.1, if the original 1-IN-3-SAT instance is completely satisfiable, then in instance $M$ it is possible to achieve a total revenue of
$$R = \frac{3K^2(s_1 + \cdots + s_N)}{\sqrt{N}} - 3K\sqrt{N} + \frac{S}{\sqrt{N}} = \frac{(9K^2 + 1)S}{\sqrt{N}} - 3K\sqrt{N}$$
and if the original 1-IN-3-SAT instance is not satisfiable then the maximum possible revenue in instance $M$ is at most
$$R' \leq \frac{9K^2 S}{\sqrt{N}} - 3K\sqrt{N} + \frac{S-1}{\sqrt{N}} + \frac{3}{K} \cdot \frac{1}{\sqrt{N}} \leq R - \frac{1}{2\sqrt{N}}$$
where the last inequality holds as long as $K \geq 6$. $\qquad\square$

*Proof.* Let $\tilde{w}$ be the optimal solution for data $(\tilde{x}^t, \tilde{v}^t)_{t=1..m}$. We will construct a vector $w$ such that:
$$\sum_t \text{REV}(\langle w, \bar{x}^t \rangle; \bar{v}^t) \geq \sum_t \text{REV}(\langle \tilde{w}, \tilde{x}^t \rangle; \tilde{v}^t) - O(\delta m)$$

Construct a vector $w$ such that $w_1 = (1 - 3\delta)(\tilde{w}_1 - 3\delta)$ and $w_i = (1 - 3\delta)\tilde{w}_i$ for $i > 1$. We have
$$||w||_2 \leq (1 - 3\delta)(1 + 3\delta) \leq 1$$
so the solution is feasible. For each point $t$ such $0 \leq \langle \tilde{w}, \tilde{x}^t \rangle \leq v^t$ observe that:
$$\langle w, \bar{x}^t \rangle \geq (1 - 3\delta)(\tilde{w}_1 - 3\delta) + (1 - 3\delta)\langle \tilde{w}_{2..n}, \bar{x}_{2..n} \rangle$$
$$\geq (1 - 3\delta)\langle \tilde{w}, \tilde{x} \rangle - 3\delta - ||\tilde{x}_{2..n} - \bar{x}_{2..n}|| \geq \langle \tilde{w}, \tilde{x} \rangle - 7\delta$$
and that:
$$\langle w, \bar{x}^t \rangle \leq (1 - 3\delta)(\tilde{w}_1 - 3\delta) + (1 - 3\delta)[\langle \tilde{w}_{2..n}, \tilde{x}_{2..n} \rangle + \delta]$$
$$\leq (1 - 3\delta)\langle \tilde{w}, \tilde{x} \rangle - (1 - 3\delta)2\delta \leq \langle \tilde{w}, \tilde{x} \rangle - \delta \leq \tilde{v}^t - \delta \leq \bar{v}^t$$
and hence
$$\text{REV}(\langle w, \bar{x}^t \rangle; \bar{v}^t) \geq \text{REV}(\langle \tilde{w}, \tilde{x}^t \rangle; \tilde{v}^t) - 5\delta$$
Summing over all $t$ gets us the desired expression. This shows in particular that $\bar{R} - \tilde{R} \leq 5\delta m$. Since the setting is symmetric, the same proof (with roles of $\tilde{R}$ and $\bar{R}$ reversed) gives us $\tilde{R} - \bar{R} \leq 5\delta m$. $\qquad\square$

**Proof of Theorem 4.1**

*Proof of Theorem 4.1.* Let $\tilde{w}$ be the optimal solution for data $(\tilde{x}^t, \tilde{v}^t)_{t=1..m}$. We will construct a vector $w$ such that:

$$\sum_t \text{REV}(\langle w, \bar{x}^t \rangle; \bar{v}^t) \geq \sum_t \text{REV}(\langle \tilde{w}, \tilde{x}^t \rangle; \tilde{v}^t) - O(\delta m)$$

Construct a vector $w$ such that $w_1 = (1 - 3\delta)(\tilde{w}_1 - 3\delta)$ and $w_i = (1 - 3\delta)\tilde{w}_i$ for $i > 1$. We have

$$\|w\|_2 \leq (1 - 3\delta)(1 + 3\delta) \leq 1$$

so the solution is feasible. For each point $t$ such $0 \leq \langle \tilde{w}, \tilde{x}^t \rangle \leq v^t$ observe that:

$$\langle w, \bar{x}^t \rangle \geq (1 - 3\delta)(\tilde{w}_1 - 3\delta) + (1 - 3\delta)\langle \tilde{w}_{2..n}, \bar{x}_{2..n} \rangle$$
$$\geq (1 - 3\delta)\langle \tilde{w}, \tilde{x} \rangle - 3\delta - \|\tilde{x}_{2..n} - \bar{x}_{2..n}\| \geq \langle \tilde{w}, \tilde{x} \rangle - 7\delta$$

and that:

$$\langle w, \bar{x}^t \rangle \leq (1 - 3\delta)(\tilde{w}_1 - 3\delta) + (1 - 3\delta)[\langle \tilde{w}_{2..n}, \tilde{x}_{2..n} \rangle + \delta]$$
$$\leq (1 - 3\delta)\langle \tilde{w}, \tilde{x} \rangle - (1 - 3\delta)2\delta \leq \langle \tilde{w}, \tilde{x} \rangle - \delta \leq \tilde{v}^t - \delta \leq \bar{v}^t$$

and hence

$$\text{REV}(\langle w, \bar{x}^t \rangle; \bar{v}^t) \geq \text{REV}(\langle \tilde{w}, \tilde{x}^t \rangle; \tilde{v}^t) - 5\delta$$

Summing over all $t$ gets us the desired expression. This shows in particular that $\bar{R} - \tilde{R} \leq 5\delta m$. Since the setting is symmetric, the same proof (with roles of $\tilde{R}$ and $\bar{R}$ reversed) gives us $\tilde{R} - \bar{R} \leq 5\delta m$.

$\square$

**Proof of Corollary 4.3**

*Proof of 4.3.* Let $w^*$ be the solution of $\max_{\|w\|_2 \leq 1} F_{\mathcal{D}}(w)$. By the previous theorem we have:

$$F_{\mathcal{D}}(w_{\mathcal{S}}) \geq F_{\mathcal{S}}(w_{\mathcal{S}}) - O(\epsilon)$$
$$F_{\mathcal{S}}(w^*) \geq F_{\mathcal{D}}(w^*) - O(\epsilon)$$

Since $F_{\mathcal{S}}(w_{\mathcal{S}}) \geq F_{\mathcal{S}}(w^*)$ by the definition of $w_{\mathcal{S}}$ we obtain the result in the statement. $\square$

## Footnotes

[2] It is worth noticing that while the Myersonian regression problem has the assumption $\|x^t\|_2 \leq 1$, in the transformation we can have $\|x^t\|_2 \leq \sqrt{3}$. We can rescale every parameter by $\sqrt{3}$, but since constants don't matter in our analysis, we keep the slightly larger norm to keep the notation simpler.