[Reviews · NeurIPS 2020]

Review 1

Summary and Contributions: This work considers the problem of learning optimal contextual prices from a set of allocation/bid samples that would arise from a users bids within an auction. Prior work has addressed this problem significantly, and shown that when generalizable, optimal prices could be computed based on the empirical distribution provided, but have not shown how to do this efficiently without assumptions on data. The work presents an approach (Myersonian Regression) that represents more direct learning of estimated losses relating to revenue in auctions, at least relative to the benchmark of Lipcshitz smooth pricing policies. The work accomplishes this through Johnson-Lindenstrauss dimensionality reduction and sampling, then solving optimally on smaller sample. The benchmark used is the optimal 1-Lipshitz pricing policy, which due to the resulting smoothness of the policy, becomes an easier target for dimensionality reduction through Johnson-Lindenstrauss.

Strengths: The work covers formulation, hardness and approximation, with fewer distributional assumptions than prior work. The approach of moving the assumptions of smoothness from the distribution to the benchmark is interesting where either for practical concerns, there would be complexity restrictions on the actual policy or in a setting it can be shown that there is usually little loss from simplicity or smoothness assumptions on the mechanism.

Weaknesses: The terminology "Myersonian Regression" suggests a more direct tie in to Myersonian Virtual Values / amortization of expected revenue (x(v - (1-F)/f)). I do not think this approach is leveraging that amortization - if it is leveraging and I missed the connect, more discussion of its contribution should be shown. The results are direct consequences from the assumption of using as a benchmark 1-Lipschitz pricing policies (for which the leveraged Johnson-Lindenstrauss sampling approach works). However, this is a very strong assumption. The optimal auction in simple settings is often a reservation price, which is not a 1-Lipschitz pricing policy (and is never once discussed in the work). The work would be made stronger with empirical justification that optimal 1-Lipschitz pricing policies are close to optimal pricing policies in the given setting. This work claims to have fewer distributional assumptions than other work, but that is really because the distributional assumptions have been moved from the algorithm into the benchmark, without a full discussion of the benchmark performance vs the full optimal benchmark. AFTER RESPONSE: The authors addressed my (misplaced) concern over applicability of 1-Lipschitz pricing policies.

Correctness: The methods seem to make sense, I was not able to fully verify proofs.

Clarity: The paper was reasonably well written, however more connection to the themes driving results (revenue characterization and sampling) would be helpful.

Relation to Prior Work: The work clearly discusses how it differs from previous contributions.

Reproducibility: Yes

Additional Feedback:


Review 2

Summary and Contributions: This paper presents the Myersonian loss function (based on Myerson price) and show that finding the optimal hypothesis exactly is NP-hard, but present a PTAS yields an additive approximation of optimal revenue. Depending on a normalization constraint added to the ERM problem (MR vs UMR), the authors show stability of (normalized) MR solutions and possible instability of UMR solutions. Note: Throughout, I use the citation [MM14] to reference A. M. Medina and M. Mohri. Learning theory and algorithms for revenue optimization in second price auctions with reserve. In Proceedings of the 31st International Conference on Machine Learning (ICML-14), pages 262–270, 2014a

Strengths: The presentation of a PTAS (and lack of FPTAS) to learn the approximate Myerson price (MP) is a strong result, and contributes to a seemingly limited literature on learning MP in a learning theory setting. Moreover, the work, in showing NP-hardness yet giving a PTAS by a JL projection for learning Myerson price gives a pretty tight characterization on the difficulty of the problem for a thorough study of the problem. - Having the clear explanation of ``the quadrants'' of online vs offline and contextual vs noncontextual settings was very helpful for me to situate this work.

Weaknesses: SUGGESTIONS FOR IMPROVING SCORE -------------------------------- - Develop and motivate a bit further the construction of the revenue function at the bottom of page 1 and if surrogates could be used. - Flesh out the literature review and spend more time contextualizing the results. - Share how the NP-hardness result is surprising if it is. - The broader impacts mentions pricing for online marketplaces; mentioning this in the motivation for *why* one might want to learn the Myerson price would strength the motivation of the paper. ================= AFTER RESPONSE ================= - Thank you for the elaboration re: the contextualization of the hardness result.

Correctness: In reviewing the proofs, I did not find any inaccuracies.

Clarity: Overall, the paper was well-written, though some of the emphasis could be reframed to focus on the perhaps more surprising results of the existence of PTAS for predicting Myersonian price. CLARIFICATIONS -------------------------------- - The origin of some constants in proofs is unintuitive, and it is unclear if they are arbitrary. (ex: exp(5), 10\epsilon) - Aren't discontinuous losses almost always NP-hard to optimize? In that sense, the result seems unsurprising. - There is a small difference between your revenue function (above line 25) and the revenue function in [MM14] Equation 1, namely that their v is in \reals^2. Can you speak to this? - Is there a consistent surrogate that makes sense in this context? ([MM14] uses a convex surrogate, but it is seemingly not consistent, see [MM14] Fig 1b.) ================= AFTER RESPONSE ================= -The use of constants for simplifying analysis makes sense; I just wasn't sure if there was something in the analysis I was missing in my reading. - Emphasizing the beauty of the hardness result might be helpful to readers, though I understand the difficulty given space constraints.

Relation to Prior Work: It took an external review of [MM14] for me to grasp how this paper differs. In the contextual offline revenue maximization setting, [MM14] is the only paper discussed. It is unclear if this is because this is the only other paper that has studied this setting, but given it was the only reference for the paper's setting, a more thorough review and comparison is needed.

Reproducibility: Yes

Additional Feedback: - Figure 1 would be very helpful to have in the body of the paper. IMPRESSIONS -------------------------------- The primary contribution of this work, it seems is the PTAS, since [MM14] shows the target loss is discontinuous, and therefore it would seem hard to optimize, making the first result seemingly unsurprising. That being said, the PTAS (and lack of FPTAS) are both interesting results worth publication on their own.


Review 3

Summary and Contributions: This paper studies Myersonian regression, which a variant of linear regression. The input is a set of samples representing the bids of a buyer for different products. The goal is to compute the (Myerson) pricing policy which maximizes the expected revenue, for any possible distribution from which the samples could be drawn from. The authors study the complexity of this problem: they show that is hard, there is no FPTAS, and present a PTAS that obtains an additive approximation of the optimal possible revenue, under normalization assumptions.

Strengths: The paper is definitely relevant to the NeurIPS community, and is almost complete in terms of sets of results: the authors have proved both the hardness of the problem (which was missing from previous work) and also presented a PTAS for it.

Weaknesses: A concrete example/applications is missing to motivate the study of the problem. Also, the assumption that the samples come from truthful auctions seem quite restrictive (since auctions used in practice are typically not truthful). The authors do mention this as a future direction though.

Correctness: The methods seem to be correct as far I could check.

Clarity: The paper is adequately written.

Relation to Prior Work: The authors thoroughly discuss the related literature and where this paper stands in terms of previous work.

Reproducibility: No

Additional Feedback: You do not need to write ||x||_2 for the 2-norm, ||x|| is enough; 2-norm is the default one for the length notation. Post rebuttal: Thank you for your response. My opinion remains the same.

[Author Response · NeurIPS 2020]

We thank all the reviewers for their detailed and insightful feedback.

**Reviewer 1:**

We would like to clarify that our setting does include "reservation price" pricing policies; in fact, they are the principal
focus of our paper. Specifically, the goal of our paper is to learn a good linear reservation pricing policy; a policy which
examines the features of the item for sale, generates a reservation price from those features, and runs a second-price
auction with this price as a reserve (in the case of a single bidder, this reduces to posting a single price for the item).
See Figure 1 in the appendix. The 1-Lipshitzness constraint is not a constraint on the function determining how much a
bidder pays given their bid (which indeed would eliminate many natural mechanisms), but rather a constraint on the
function determining the reservation price as a function of the features.

In particular, note that our benchmark contains the optimal mechanism in the simple (non-contextual) setting.

**Reviewer 2:**

We thank the review for the suggestions to improve the paper which we plan to implement in the revision: we will
extend the discussion on motivation and add more context to the results. Below we respond to questions asked:

- 14 • Why is hardness surprising? Indeed the discontinuity of the loss function hints at hardness, but it has been
  15 open for a while and in fact we are not aware of any straightforward reduction. Even though hardness is not
  16 surprising it wasn't at all clear where the problem was situated in the hardness-spectrum,i.e. it is APX-hard or
  17 constant approx or PTAS or FPTAS?

- 18 • Constants in the proofs: constants are somewhat arbitrary as we don't try to optimize them (instead we simply
  19 pick constants that are convenient for the analysis). We choose to write explicit constants that satisfy the
  20 necessary inequalities (instead of just using $O(\cdot)$) since the conditions for both JL and the hardness reduction
  21 are somewhat delicate and often require checking both upper and lower bounds.

- 22 • We discuss the loss of [MM14] in the last paragraph of Section 4 in our paper. We call it the SPA loss. It takes
  23 two bids $(v_1, v_2)$ and returns the revenue of a second price auction with reserve. Throughout our paper we
  24 study the simpler pricing loss which is equivalent to SPA with $v_2 = 0$. However in the end of Section 4 we
  25 note that all our results extend to SPA and any other lower-semi-Lipschitz loss function.

- 26 • The question of consistent surrogates is an interesting one. The existence of convex consistent surrogates is
  27 ruled-out by our hardness result. One could still search for consistent surrogates without poly-time guarantees
  28 but that are smooth enough to be tractable in practice. Since our focus here was to obtain provable guarantees,
  29 we opted for working with the loss function directly.

30 **Reviewer 3:**

31 Thanks for the suggestions to improve the paper. We plan to implement those in the revision. We will add a discussion
32 in future directions on how to extend this work to non-truthful auctions (such as first-price and GSP) as well as more
33 motivation for the truthful case (e.g single slot second price and VCG auctions often used in display ads). We will
34 expand the discussion on the future direction of strategic response in the conclusion.

[Meta-Review · NeurIPS 2020]

All reviewers agree that this is a strong technical contribution and all major concerns were addressed in the rebuttal.